# CP² skyrmions and skyrmion crystals in realistic quantum magnets

Hao Zhang ®[1,2,3] ✉, Zhentao Wang ®[1,4,6], David Dahlbom ®[1], Kipton Barros[3] & Cristian D. Batista ®[1,5] ✉

Magnetic skyrmions are nanoscale topological textures that have been recently observed in different families of quantum magnets. These objects are called $CP^1$ skyrmions because they are built from dipoles−the target manifold is the 1D complex projective space, $CP^1 \cong S^2$. Here we report the emergence of magnetic $CP^2$ skyrmions in a realistic spin-1 model, which includes both dipole and quadrupole moments. Unlike $CP^1$ skyrmions, $CP^2$ skyrmions can also arise as metastable textures of quantum paramagnets, opening a new road to discover emergent topological solitons in non-magnetic materials. The quantum phase diagram of the spin-1 model also includes magnetic field-induced $CP^2$ skyrmion crystals that can be detected with regular momentum- (diffraction) and real-space (Lorentz transmission electron microscopy) experimental techniques.

Lord Kelvin's vision of the atom as a vortex in ether[1] inspired Skyrme[2,3] to explain the origin of nucleons as emergent topologically non-trivial configurations of a pion field described by a $3+1$ dimensional O(4) non-linear $\sigma$-model. In the modern language, these "skyrmions" are examples of topological solitons, and Skyrme's model has become the prototype of a classical theory that supports these solutions. Besides its important role in high-energy physics and cosmology, Skyrme's model also led to important developments in other areas of physics. For instance, the baby Skyrme model[4–6] (planar reduction of the non-linear $\sigma$-model), which is an extension of the Heisenberg model[4,5,7], has baby skyrmion solutions in the presence of a chiral symmetry-breaking Dzyaloshinskii–Moriya (DM) interaction[8–11].

Periodic arrays of magnetic skyrmions and single skyrmion metastable states were originally observed in chiral magnets, such as MnSi, $Fe_{1−x}Co_xSi$, FeGe, and $Cu_2OSeO_3$[12–16]. This discovery sparked the interest of the community at large and spawned efforts in multiple directions. Identifying realistic conditions for the emergence of novel magnetic skyrmions is one of the main goals of modern condensed matter physics. Novel mechanisms are usually accompanied by new properties. For instance, while skyrmions of chiral magnets have a

fixed vector chirality, this is still a degree of freedom in centrosymmetric materials, such as $BaFe_{1−x−0.05}Sc_xMg_{0.05}O_{19}$, $La_{2−2x}Sr_{1+2x}Mn_2O_7$, $Gd_2PdSi_3$, and $Gd_3Ru_4Al_{12}$[17–23], where skyrmions arise from frustration, i.e., from competing exchange or dipolar interactions[24–30].

The target manifold of the above-mentioned planar baby skyrmions is $S^2 \cong CP^1$, i.e., the usual 2D sphere, corresponding to normalized dipoles. More generally, one may consider the complex projective space $CP^{N−1}$ that represents the normalized $N$-component complex vectors, up to an irrelevant complex phase. The topologically distinct, smooth mappings from the base manifold $S^2$ (2D sphere $\cong$ compactified plane) to the target manifold $CP^{N−1}$ can be labeled by the integers: $\Pi_2(CP^{N−1}) = \mathbb{Z}$. This homotopy group suggests generalizations of the planar Skyrme's model to $N > 2$, such as the $CP^2$ nonlinear $\sigma$-model[31–33] and in the Faddeev-Skyrme type model[34,35]. In condensed matter physics, chiral $CP^2$ skyrmion configurations induced by fluctuations or quenching the system through a phase transition were proposed in the context of three-band superconductors with broken time-reversal symmetry[36–38]. In recent work, Akagi et al. considered the SU(3) version of the Heisenberg model with a DM interaction, whose continuum limit becomes a gauged $CP^2$ nonlinear $\sigma$-model with a

[1]Department of Physics and Astronomy, The University of Tennessee, Knoxville, TN 37996, USA. [2]Materials Science and Technology Division, Oak Ridge National Laboratory, Oak Ridge, TN 37831, USA. [3]Theoretical Division and CNLS, Los Alamos National Laboratory, Los Alamos, NM 87545, USA. [4]School of Physics and Astronomy, University of Minnesota, Minneapolis, MN 55455, USA. [5]Quantum Condensed Matter Division and Shull-Wollan Center, Oak Ridge National Laboratory, Oak Ridge, TN 37831, USA. [6]Present address: Center for Correlated Matter and School of Physics, Zhejiang University, Hangzhou 310058, China. ✉e-mail: zhang_h@lanl.gov; cbatist2@utk.edu

background uniform gauge field[39]. An attractive aspect of this model is that it admits analytical solutions by the application of techniques developed for the gauged non-linear $\sigma$-model. However, it may be challenging to find materials described by this model because SU(3) can only be an accidental symmetry of the spin–spin interactions of real quantum magnets, and Hamiltonians that do exhibit SU(3)-invariance contain unrealistically strong biquadratic terms. In insulating magnets, biquadratic interactions are typically much smaller than bilinear interactions because they are of higher order in the small parameter that leads to the emergence of magnetic moments (localized electrons) in real materials (e.g., the ratio $t/U$ between the typical hopping amplitude, $t$, and the on-site Hubbard repulsion, $U$, in the case of Mott insulators). Similar limitations apply to other works that study skyrmion solutions of the bilinear-biquadratic spin one model[40–43].

The main purpose of this work is to demonstrate that exotic $CP^2$ skyrmions readily emerge in a simple and realistic spin-1 ($N = 3$) model and its natural extensions. In other words, we propose that these magnetic textures could likely be observed in real materials. Remarkably, isolated $CP^2$ skyrmions can either be metastable states of a quantum paramagnet (QPM) or a fully polarized (FP) ferromagnet. Unlike the "usual" $CP^1$ magnetic skyrmions, the dipolar field of the metastable $CP^2$ skyrmions of quantum paramagnets vanishes away from the skyrmion core. Moreover, the application of an external magnetic field to the QPM induces stable triangular crystals of $CP^2$ skyrmions in the field interval that separates the QPM from the FP state.

## Model

To illustrate the basic ideas, we consider a minimal spin-1 model defined on the triangular lattice (TL):

$$\hat{\mathcal{H}} = \sum_{\langle i,j\rangle} J_{ij}\left(\hat{S}_i^x\hat{S}_j^x + \hat{S}_i^y\hat{S}_j^y + \Delta\hat{S}_i^z\hat{S}_j^z\right) - h\sum_i \hat{S}_i^z + D\sum_i \left(\hat{S}_i^z\right)^2. \quad (1)$$

The first term includes an easy-axis ferromagnetic (FM) nearest-neighbor exchange interaction $J_1 < 0$ and a second-nearest-neighbor antiferromagnetic (AFM) exchange $J_2 > 0$. For simplicity, we assume that the exchange anisotropy, defined by the dimensionless parameter $\Delta > 1$, is the same for both interactions. The second and third terms represent the Zeeman coupling to an external field and an easy-plane single-ion anisotropy ($D > 0$). $\hat{\mathcal{H}}$ is invariant under the space group of the TL and the U(1) group of global spin rotations along the field axis. We will adopt $|J_1|$ as the unit of energy (i.e. $J_1 = -1$).

To study the properties of the skyrmion solutions of Eq. (1), it is helpful to consider the classical limit first. In doing so, we follow the existing literature on topological solitons, which are inherently classical objects. There is, moreover, good reason to expect that quantum fluctuations are not relevant to the present study. Experimentally, there is existing evidence of spin-1 triangular materials that exhibit semi-classical spiral orderings due to competing ferromagnetic and antiferromagnetic exchange interactions[44]. Some of these are discussed in Section "Disussion". Furthermore, the results that will be developed will remain unaltered for a simple 3D extension of the current model, achieved by vertically stacking triangular layers with ferromagnetic interlayer coupling. The larger coordination number of the 3D model and the long wavelength nature of the ordered states both act to reduce quantum fluctuations, further justifying the classical approximation.

It is important to note, however, that there are subtleties in formulating the appropriate classical limit[45,46]. The traditional classical limit is based on SU(2) coherent states, which retain only the spin dipole expectation value and produces the Landau–Lifshitz spin dynamics. This approach is adequate for modeling systems with weak single-ion anisotropy $D \ll |J_1|$. To classically model systems in the regime $D \gtrsim |J_1|$, however, it is necessary to retain more structure from

the quantum spin-1 states, which live in a local Hilbert space of dimension $N = 3$. Specifically, our classical limit will assume that the many-body quantum state is a direct product of SU(3) coherent states[45–52]:

$$|\mathbf{Z}\rangle = \otimes_j |\mathbf{Z}_j\rangle \quad \text{with} \quad |\mathbf{Z}_j\rangle = \sum_a Z_j^a |x^a\rangle_j, \quad (2)$$

where $\mathbf{Z}_j = (Z_j^1, Z_j^2, Z_j^3)^{\mathsf{T}}$ is a complex vector of unit length and $\{|x^1\rangle_j, |x^2\rangle_j, |x^3\rangle_j\}$ is an orthonormal basis for the local Hilbert state on-site $j$.

Local physical operators are represented by Hermitian matrices that act on SU(3) coherent states. The space of $3 \times 3$ traceless, Hermitian matrices comprises the fundamental representation of the $\mathfrak{su}(3)$ Lie algebra. A basis $\hat{T}^\mu$ ($\mu = 1, ..., 8$) for this space is characterized by the commutation relations,

$$\left[\hat{T}_j^\eta, \hat{T}_j^\mu\right] = if_{\eta\mu\nu}\hat{T}_j^\nu, \quad (3)$$

where we are using the convention of summation over repeated Greek indices. We may additionally impose an orthonormality condition

$$\mathrm{Tr}\left(\hat{T}_j^\alpha \hat{T}_j^\beta\right) = 2\delta_{\alpha\beta}. \quad (4)$$

It is conventional to define the structure constants as $f_{\eta\mu\nu} = -\frac{i}{2}\mathrm{Tr}\left(\lambda_\eta[\lambda_\mu, \lambda_\nu]\right)$, where $\lambda_\mu$ are the Gell–Mann matrices.

The spin dipole operators $\hat{\mathbf{S}}_j = (\hat{S}_j^x, \hat{S}_j^y, \hat{S}_j^z)^{\mathsf{T}}$ acting on site $j$ are generators for a spin-1 representation of SU(2). It is possible to formulate generators of SU(3) as polynomials of these spin operators,

$$\begin{pmatrix}\hat{T}_j^7 \\ \hat{T}_j^5 \\ \hat{T}_j^2\end{pmatrix} = -\begin{pmatrix}\hat{S}_j^x \\ \hat{S}_j^y \\ \hat{S}_j^z\end{pmatrix}, \quad \begin{pmatrix}\hat{T}_j^3 \\ \hat{T}_j^8 \\ \hat{T}_j^1 \\ \hat{T}_j^4 \\ \hat{T}_j^6\end{pmatrix} = \begin{pmatrix}-\left(\hat{S}_j^x\right)^2 + \left(\hat{S}_j^y\right)^2 \\ \frac{1}{\sqrt{3}}\left[3\left(\hat{S}_j^z\right)^2 - \hat{\mathbf{S}}_j^2\right] \\ \hat{S}_j^x\hat{S}_j^y + \hat{S}_j^y\hat{S}_j^x \\ -\hat{S}_j^z\hat{S}_j^x - \hat{S}_j^x\hat{S}_j^z \\ \hat{S}_j^y\hat{S}_j^z + \hat{S}_j^z\hat{S}_j^y\end{pmatrix}, \quad (5)$$

where $T_j^{7,5,2}$ are the dipolar components of the spin-1 degree of freedom, while the other five generators are the quadrupolar components. Here we have adopted the notation and conventions of ref. 39 to make closer contact with the literature on high-energy physics. (Our definitions for $\hat{S}^x$ and $\hat{S}^z$ differ from these two in ref. 39 by a minus sign).

Let $|1\rangle_j$, $|0\rangle_j$, and $|\bar{1}\rangle_j$ denote the normalized eigenstates of $\hat{S}_j^z$, with eigenvalues, 1, 0 and $-1$, respectively. In the Cartesian basis,

$$|x^1\rangle_j = \frac{i\left[|1\rangle_j - |\bar{1}\rangle_j\right]}{\sqrt{2}}, |x^2\rangle_j = \frac{\left[|1\rangle_j + |\bar{1}\rangle_j\right]}{\sqrt{2}}, |x^3\rangle_j = -i|0\rangle_j, \quad (6)$$

the SU(3) generators given in Eq. (5) are the Gell−Mann matrices:

$$\left\langle x_j^a|\hat{T}_j^\mu|x_j^b\right\rangle = (\lambda_\mu)_{ab} \quad \mu = 1, 2, \dots, 8. \quad (7)$$

The orbit of coherent states $|\mathbf{Z}_j\rangle$ is obtained by applying SU(3) transformations to the highest weight state $|1\rangle_j$[45]: $|\mathbf{Z}_j\rangle = \hat{U}_j|1\rangle_j$. Since the global phase is a gauge degree of freedom, the orbit of physical SU(3) coherent states is $S^5/S^1 \cong CP^2$. The "SU(3) classical limit" of the spin Hamiltonian (1) is obtained by replacing the Hamiltonian operator $\hat{\mathcal{H}}$ with its expectation value

$$\mathcal{H} \equiv \langle \mathbf{Z}|\hat{\mathcal{H}}|\mathbf{Z}\rangle, \quad (8)$$

after rewriting $\hat{\mathcal{H}}$ in terms of SU(3) spin components,

$$\hat{\mathcal{H}} = \sum_{\langle i,j \rangle} I^{\mu}_{ij} \hat{T}^{\mu}_i \hat{T}^{\mu}_j - \sum_i B^{\mu} \hat{T}^{\mu}_i, \qquad (9)$$

where $I^{\mu}_{ij} = J_{ij}(\Delta\delta_{\mu,2} + \delta_{\mu,5} + \delta_{\mu,7})$ and $B^{\mu} = (-h\delta_{\mu,2} - D\delta_{\mu,8}/\sqrt{3})$. Because the direct product form of Eq. (2), $\mathcal{H}$ can be expressed as a function of the "color field"

$$n^{\mu}_j \equiv \left\langle \mathbf{Z}_j | \hat{T}^{\mu}_j | \mathbf{Z}_j \right\rangle = (\lambda_{\mu})_{ab} \bar{Z}^a_j Z^b_j, \qquad (10)$$

which satisfies the constraints

$$n^{\mu}n^{\mu} = \frac{4}{3}, \quad n^{\mu} = \frac{3}{2} d_{\mu\nu\eta} n^{\nu} n^{\eta}, \qquad (11)$$

where $d_{\mu\nu\eta} = \frac{1}{4}\mathrm{Tr}(\lambda_{\mu}\{\lambda_{\nu},\lambda_{\eta}\})$. This in turn leads to the Casimir identity: $d_{mpq}n^m n^p n^q = \frac{8}{9}$. In terms of this color field, we can express

$$\mathcal{H} = \sum_{\langle i,j \rangle} I^{\mu}_{ij} n^{\mu}_i n^{\mu}_j - \sum_i B^{\mu} n^{\mu}_i. \qquad (12)$$

To avoid an explicit use of the structure constants ($f_{\eta\mu\nu}$), we introduce an equivalent formulation of the problem using the operator field $\mathfrak{n}_j = n^{\mu}_j \lambda_{\mu}$. Topological soliton solutions of the color field become well-defined in the continuum (long wavelength) limit, where the Hamiltonian can be approximated by

$$\mathcal{H} \simeq \int \mathrm{d}r^2 \left[ -\frac{\mathcal{I}^{\mu}_1}{2}(\nabla n^{\mu})^2 + \frac{\mathcal{I}^{\mu}_2}{2}(\nabla^2 n^{\mu})^2 - \mathcal{B}^{\mu} n^{\mu} \right], \qquad (13)$$

where $\nabla$ denotes the spatial gradient operator. The coupling constants can be expressed in terms of the parameters of the lattice model (9):

$$\mathcal{I}^{\mu}_1 = \frac{3}{2}(I^{\mu}_1 + 3I^{\mu}_2), \quad \mathcal{I}^{\mu}_2 = \frac{3}{32}(I^{\mu}_1 + 9I^{\mu}_2),$$
$$\mathcal{B}^{\mu} = B^{\mu} - 3(\Delta - 1)(J_1 + J_2)\delta_{\mu,8}. \qquad (14)$$

Eq. (13) corresponds to an anisotropic $CP^2$ model. For skyrmion solutions, the base plane $\mathbb{R}^2$ can be compactified to $S^2$ because the color field takes a constant value $n_{\infty}$ at spatial infinity. These spin textures can then be characterized by the topological charge of the mapping $\mathfrak{n} : \mathbb{R}^2 \sim S^2 \mapsto CP^2$:

$$C = -\frac{i}{32\pi} \int \mathrm{dxdy}\,\varepsilon_{jk} \mathrm{Tr}\left( \mathfrak{n}\left[ \partial_j \mathfrak{n}, \partial_k \mathfrak{n} \right] \right). \qquad (15)$$

For the lattice systems of interest, the $CP^2$ skyrmion charge can be computed after interpolating the color fields on nearest-neighbor sites $\mathfrak{n}_j$ and $\mathfrak{n}_k$ along the $CP^2$ geodesic:

$$C = \sum_{\triangle_{jkl}} \rho_{jkl} = \frac{1}{2\pi} \sum_{\triangle_{jkl}} \left( \gamma_{jl} + \gamma_{lk} + \gamma_{kj} \right), \qquad (16)$$

where $\triangle_{jkl}$ denotes each oriented triangular plaquette of nearest-neighbor sites $j \to k \to l$ and $\gamma_{kj} = \arg[\langle \mathbf{Z}_k | \mathbf{Z}_j \rangle]$ is the Berry connection on the bond $j \to k$ (see Supplemental Material). We emphasize that the color field formalism just discussed is fully equivalent to the formalism based on coherent states. In particular, it is straightforward to show that the operator representation of the color field may be expressed as $\mathfrak{n}_j = |\mathbf{Z}_j\rangle\langle\mathbf{Z}_j| - \mathbb{1}/3$, in which form it becomes clear that both $\mathfrak{n}$ and the coherent state $|\mathbf{Z}_j\rangle$ provide equivalent representations of the same classical state[53,54].

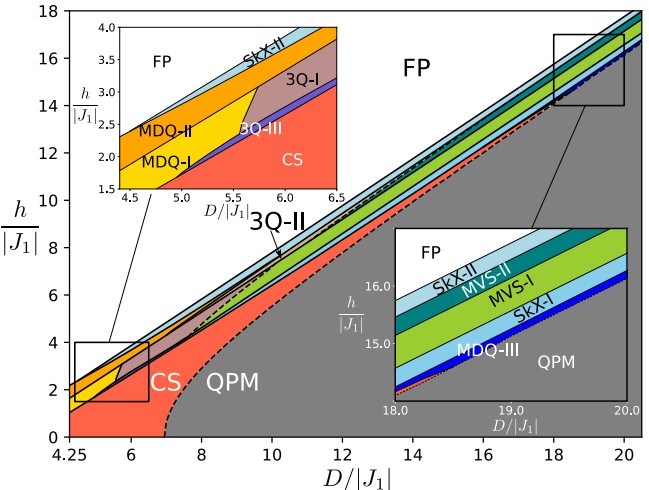

**Fig. 1 | $T = 0$ phase diagram of the classical Hamiltonian $\mathcal{H}$ as a function of the single-ion anisotropy $D$ and the external field $h$, for $J_2/|J_1| = 2/(1+\sqrt{5})$ and $\Delta = 2.6$.** The two insets show the phases for small-$D$ and large-$D$, respectively. The solid (dashed) lines indicate 1st- (2nd-) order phase transitions.

## Results

### Phase diagram

The $T = 0$ phase diagram (Fig. 1) is obtained by numerically minimizing the classical spin Hamiltonian $\mathcal{H}$ (12) in the $4L^2$-dimensional phase space of a magnetic cell of $L \times L$ spins (see the "Methods" section). The shape and size of this unit cell is dictated by the symmetry-related magnetic ordering wave vectors $\mathbf{Q}_{\nu}$ ($\nu = 1, 2, 3$) (see Fig. 2a, b), which are determined by minimizing the exchange interaction in momentum space: $J(\mathbf{q}) = \sum_{jl} J_{jl} e^{i\mathbf{q}\cdot(\mathbf{r}_j - \mathbf{r}_l)}$. The ratio between both exchange interactions, $J_2/|J_1| = 2/(1+\sqrt{5})$ is tuned to fix the magnitude of the ordering wave vectors, $|\mathbf{Q}_{\nu}| = |\mathbf{b}_1|/5^{27}$, corresponding to a magnetic unit cell of linear size $L = 5$. As we will see later, the relevant qualitative aspects of the phase diagram do not depend on the particular choice of the model (see the section "Large-$D$ limit"). The three ordering wave vectors, which are related by the $C_6$ symmetry of the TL, are parallel to the $\Gamma$-$M_{\nu}$ directions (denoted in Fig. 2).

The resulting phase diagram shown in Fig. 1 includes multiple magnetically ordered phases between the FP phase and the QPM phase, where every spin is in the $|0\rangle$ state. For $D \gg |J_1|$, these phases include two field-induced $CP^2$ skyrmion crystals (SkX-I and SkX-II), separated by two modulated vertical spiral phases (MVS-I and MVS-II), where the polarization plane of the spiral is parallel to the $c$-axis and the magnitude of the dipole moment is continuously suppressed as the moment rotates from $\hat{z}$ to $-\hat{z}$ directions. The spiral phases have the same symmetry and are separated by a first-order metamagnetic transition. As shown in Fig. 2a, the $CP^2$ skyrmions of the SkX-I crystal have dipole moments that evolve continuously into the purely nematic state ($|0\rangle$) as they move away from the core. Conversely, Fig. 2b shows that the spins in the SkX-II phase have a strong quadrupolar character (the small dipolar moment is completely suppressed in the large $D/|J_1|$ limit) at the skyrmion core, and evolve continuously into the magnetic state $|1\rangle$ as they move away from the core. The $CP^2$ skyrmion density distribution $\rho_{jkl}$ is also indicated with colored triangular plaquettes in Fig. 2a, b for SkX-I and SkX-II, respectively. As shown in the inset of Fig. 1, phase SkX-II extends down to $D/|J_1| \simeq 5$, while phase SkX-I disappears near $D/|J_1| \simeq 8$.

New competing orderings appear in the intermediate $D/|J_1|$ region. In particular, a significant fraction of the phase diagram is occupied by the so-called canted spiral (CS) phase,

$$|\mathbf{Z}_j\rangle = \cos\theta|0\rangle_j + e^{i\mathbf{Q}\cdot\mathbf{r}_j}\sin\theta\cos\phi|1\rangle_j + e^{-i\mathbf{Q}\cdot\mathbf{r}_j}\sin\theta\sin\phi|\bar{1}\rangle_j, \qquad (17)$$

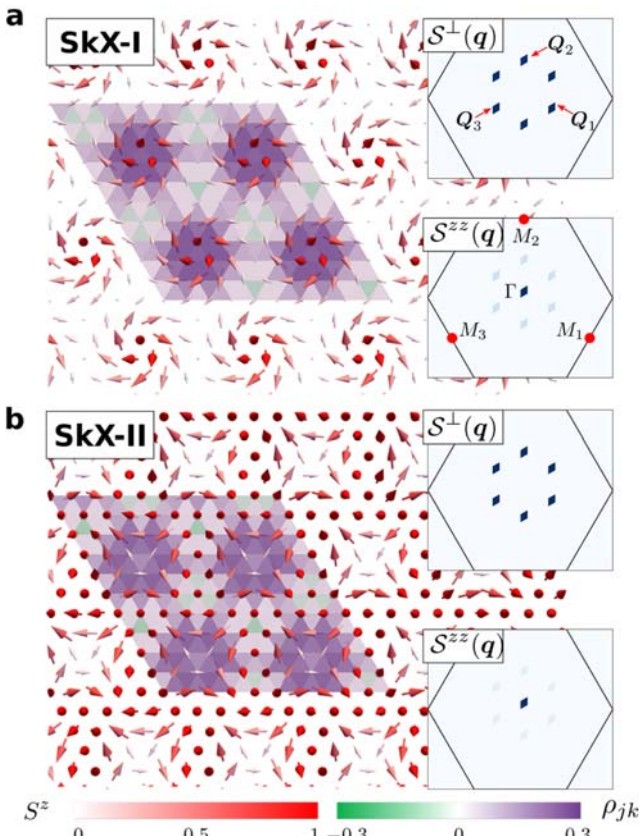

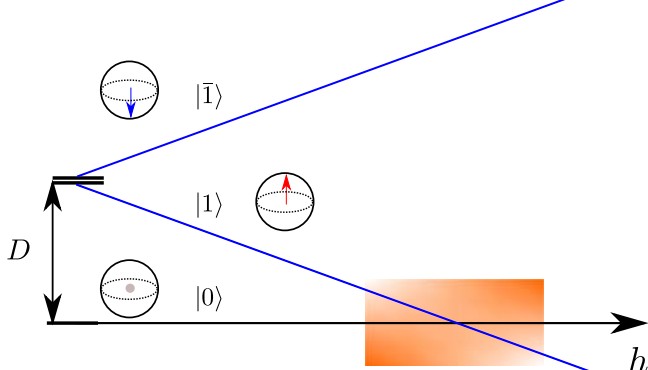

**Fig. 3 | Spectrum of the single-ion model $\hat{\mathcal{H}}_{\mathrm{SI}} = D(\hat{s}^z)^2 - h\hat{s}^z$.** The shaded region denotes the effective regime with a quasi-degenerate doublet: $\{|0\rangle, |1\rangle\}$.

low-energy Hamiltonian,

$$\hat{\mathcal{H}}_{\mathrm{eff}} = \sum_{\langle i,j \rangle} \tilde{J}_{ij} \left( \hat{s}_i^x \hat{s}_j^x + \hat{s}_i^y \hat{s}_j^y + \tilde{\Delta} \hat{s}_i^z \hat{s}_j^z \right) - \tilde{h} \sum_i \hat{s}_i^z. \tag{19}$$

The pseudo-spin-1/2 operators are the projection of the original spin operators into the low-energy subspace $\mathcal{S}_0$ generated by the quasi-degenerate doublet $\{|0\rangle_j, |1\rangle_j\}$ (see Fig. 3):

$$\hat{s}_j^z = \mathcal{P}_0 \hat{S}_j^z \mathcal{P}_0 - \frac{1}{2}, \quad \hat{s}_j^{\pm} = \frac{\mathcal{P}_0 \hat{S}_j^{\pm} \mathcal{P}_0}{\sqrt{2}}, \tag{20}$$

where $\mathcal{P}_0$ is the projection operator of the low-energy subspace. Importantly, the first state of the doublet has a net quadrupolar moment but no net dipole moment, $\langle 0|\hat{S}_j|0\rangle_j = 0$, while the second state maximizes the dipole moment along the $\hat{z}$-direction $\langle 1|\hat{S}_j|1\rangle_j = \hat{z}$. This means that three pseudo-spin operators generate an SU(2) subgroup of SU(3) different from the SU(2) subgroup of spin rotations.

$\hat{\mathcal{H}}_{\mathrm{eff}}$ represents an effective triangular easy-axis XXZ model with effective exchange, anisotropy and field parameters $\tilde{J}_{ij} = 2J_{ij}$, $\tilde{\Delta} = \frac{\Delta}{2}$ and $\tilde{h} = h - D - 3\Delta(J_1 + J_2)$, respectively. This model is known to exhibit a field-induced $CP^1$ SkX phase[25,27] on a lattice for fixed choice of $J_1$ and $J_2$. Further study has demonstrated that the full field-anisotropy phase diagram remains qualitatively the same upon approaching the long wavelength limit of $J_2 \to \frac{1}{3}|J_1|$, the Lifshitz point where the ordering wave vectors go to zero[26]. It follows that lattice effects do not alter the qualitative features of the phase diagram for wavelengths at least as long as that set by the $J_1$ and $J_2$ examined here. In other words, these results do not depend on a fine-tuning of exchange parameters. Indeed, the continuum model for Eq. (20) matches the universal Hamiltonian presented in[26],

$$\mathcal{H}_{\mathrm{eff}} \simeq \int d\mathbf{r}^2 \left[ -\frac{\mathcal{J}_1^{\eta}}{2}(\nabla \tilde{n}^{\eta})^2 + \frac{\mathcal{J}_2^{\eta}}{2}\left(\nabla^2 \tilde{n}^{\eta}\right)^2 - \tilde{\mathcal{B}} \tilde{n}_z + \tilde{\mathcal{D}} \tilde{n}_z^2 \right], \tag{21}$$

where $\eta = x, y, z$ denotes the three components of the unit vector field $\tilde{\mathbf{n}}$ ($|\tilde{\mathbf{n}}| = 1$), and

$$\begin{aligned} \mathcal{J}_1^{\eta} &= \frac{3s^2}{2}(\tilde{J}_1 + 3\tilde{J}_2)[1 + (\tilde{\Delta} - 1)\delta_{\eta z}], \\ \mathcal{J}_2^{\eta} &= \frac{3s^2}{32}(\tilde{J}_1 + 9\tilde{J}_2)[1 + (\tilde{\Delta} - 1)\delta_{\eta z}] \\ \tilde{\mathcal{B}} &= s\tilde{h}, \quad \tilde{\mathcal{D}} = 3s^2(\tilde{\Delta} - 1)(\tilde{J}_1 + \tilde{J}_2), \end{aligned} \tag{22}$$

where $s = 1/2$. Although the target manifold of this theory is $CP^1$ (orbit of SU(2) coherent states that belong $\mathcal{S}_0$), we must keep in mind that $\hat{\mathcal{H}}_{\mathrm{eff}}$ describes the large $D/|J_1|$ limit where the $CP^2$ skyrmions of the

$S^z$ 0 0.5 1 −0.3 0 0.3 $\rho_{jkl}$

**Fig. 2 | The $CP^2$ skyrmion crystal phases. a, b** Real space distribution of the dipolar sector of the $CP^2$ skyrmion crystals SkX-I and SkX-II. The length of the arrow represents the magnitude of the dipole moment of the color field $|\langle \hat{\mathbf{S}}_j \rangle| = \sqrt{(n_j^7)^2 + (n_j^5)^2 + (n_j^2)^2}$. The color scale of the arrows indicates $\langle \hat{S}_j^z \rangle = -n_j^2$. The insets display the static spin structure factors $\mathcal{S}^{\perp}(\mathbf{q}) = n_{\mathbf{q}}^7 n_{-\mathbf{q}}^7 + n_{\mathbf{q}}^5 n_{-\mathbf{q}}^5$ and $\mathcal{S}^{zz}(\mathbf{q}) = n_{\mathbf{q}}^2 n_{-\mathbf{q}}^2$, with $\mathbf{n_q} = \sum_j e^{i\mathbf{q}\cdot\mathbf{r}_j} \mathbf{n}_j/L$. The $CP^2$ skyrmion density distribution $\rho_{jkl}$ [see Eq. (16)] is indicated by the color of the triangular plaquettes.

where $\theta$ and $\phi$ are variational parameters, and $\mathbf{Q}$ can take any values among $\{\mathbf{Q}_1, \mathbf{Q}_2, \mathbf{Q}_3\}$. Upon increasing $D$, the magnitude of the dipole moment of each spin, $|\langle \hat{\mathbf{S}}_j \rangle|$, is continuously suppressed to zero at the boundary,

$$D_c = h \sqrt{1 - \frac{4J^2(\mathbf{Q})}{h^2 + 4J^2(\mathbf{Q})}} - 2J(\mathbf{Q}) \left( 1 - \frac{2J(\mathbf{Q})}{\sqrt{h^2 + 4J^2(\mathbf{Q})}} \right), \tag{18}$$

that signals the second-order transition into the QPM phase. As shown in Fig. 1, several competing phases appear above the CS phase upon increasing $h$. These phases include three triple-$\mathbf{Q}$ spiral orderings [3$\mathbf{Q}$ I–III] with dominant weight in one of three $\mathbf{Q}$ transverse components and a staggered distribution of the $CP^2$ skyrmion density $\rho_{jkl}$ [see Eq. (16)] and three different modulated double-$\mathbf{Q}$ orderings (MDQ I–III) and two triple-$\mathbf{Q}$ orderings. All of these phases are described in detail in the supplementary information. In the rest of the paper, we will focus on the SkX phases and the single-skyrmion metastable solutions that emerge in their proximity.

**Large-$D$ limit**

The origin of the $CP^2$ skyrmion crystals can be understood by analyzing the small $|J_{ij}|/D$ regime, where $\hat{\mathcal{H}}$ can be reduced via first-order degenerate perturbation theory in $J_{ij}/D$ to an effective pseudo-spin-1/2

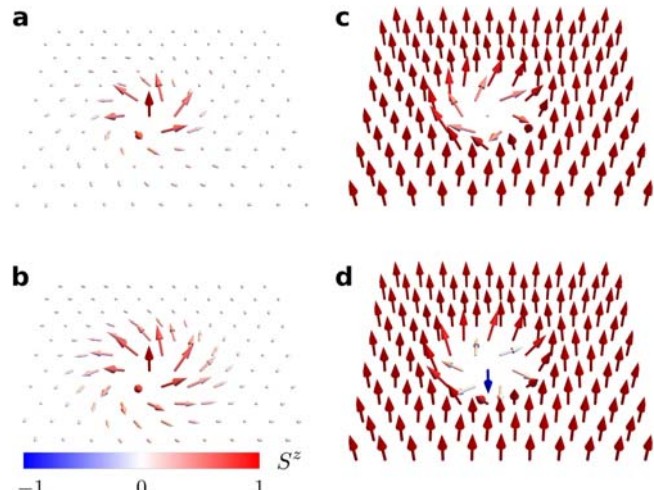

**Fig. 4 | Dipolar sector of CP² skyrmions.** The color scale indicates the value of $n_j^z$ ($\langle \hat{S}_j^z \rangle$). **a**, **b** Skyrmion excitation on top of a QPM background. **c**, **d** Skyrmion excitation on top of a fully polarized background. $J_2/|J_1| = 2/(1+\sqrt{5})$ and $\Delta = 2.6$ in (**a**), (**c**), and (**d**). $J_2/|J_1| = 2/(3+\sqrt{5})$ and $\Delta = 2.2$ in **b**. In these panels, **a** $D = 17.1|J_1|$, $H = 13|J_1|$. **b** $D = 18.3|J_1|$, $H = 14|J_1|$. **c** $D = 7|J_1|$, $H = 5|J_1|$. **d** $D = 4|J_1|$, $H = 2|J_1|$.

original spin-1 model become asymptotically close to CP¹ pseudo-spin skyrmions. In other words, the SkXs include a finite $|\bar{1}\rangle$ component for finite $D/|J_1|$, which increases upon decreasing $D$. This component, which only appears in the low-energy model when second-order corrections in $J_{ij}$ are included, is responsible for the metamagnetic transition between the MVS-I and MVS-II phases (the transition disappears in the $D \to \infty$ limit).

Since $\hat{\mathcal{H}}_{\text{eff}}(h)$ and $\hat{\mathcal{H}}_{\text{eff}}(-h)$ are related by a pseudo-time-reversal (PTR) transformation ($\hat{s}_j \to -\hat{s}_j$ on the lattice and $\tilde{n} \to -\tilde{n}$ in the continuum) their corresponding ground states are related by the same transformation. In particular, the ground state ($\tilde{\boldsymbol{n}} = \hat{\boldsymbol{z}}$) that is obtained above the saturation field ($\tilde{\mathcal{B}} > \tilde{\mathcal{B}}_{\text{sat}}$) corresponds to the FP state ($\langle \hat{\boldsymbol{S}}_j \rangle = \hat{\boldsymbol{z}}$) in the original spin-1 variables, while the ground state ($\tilde{\boldsymbol{n}} = -\hat{\boldsymbol{z}}$) below the negative saturation field ($\tilde{\mathcal{B}} < -\tilde{\mathcal{B}}_{\text{sat}}$) corresponds to the QPM phase ($|\boldsymbol{Z}_j\rangle = |0\rangle_j$). Correspondingly, the SkX induced by a positive $h$ has pseudo-spins polarized along the quadrupolar direction ($|0\rangle$) near the core of the skyrmions and parallel to the dipolar one ($|1\rangle$) at the midpoints between two cores. This explains the origin of the SkX-II crystals depicted in Fig. 2b. The negative $\mathcal{B}$ counterpart of this phase, which is obtained by applying the PTR transformation, corresponds to the SkX-I crystal shown in Fig. 2a. In this case the skyrmion cores are magnetic, while the midpoints are practically quadrupolar (they become purely quadrupolar in the large $D/|J_1|$ limit). This simple reasoning explains the origin of the novel SkX phases included in the $T = 0$ phase diagram of $\mathcal{H}$ shown in Fig. 1. The intermediate phase between the SkX-I and SkX-II ground state of $\mathcal{H}$ induced by positive and negative values of $h$ is a single-$\boldsymbol{Q}$ spiral with a polarization plane parallel to the $c$-axis known as a vertical spiral (VS). This explains the origin of the MVS-I and MVS-II phases in between the two SkX phases (the first-order transition between both phases disappears in the large-$D$ limit[25]).

**Single-skyrmion solutions**

Besides the SkX phases shown in Fig. 4, the effective field theory (21) is known to support metastable CP¹ single-skyrmion solutions beyond the saturation fields $|\tilde{\mathcal{B}}| > \tilde{\mathcal{B}}_{\text{sat}}$. The pseudo-spin variable is anti-parallel to the external field at the core and it gradually rotates towards the direction parallel to the field upon moving away from the center. Interestingly, this pseudo-spin texture translates into metastable single-skyrmion solutions of the QPM phase that have a magnetic core and a nematic periphery, as it is shown in Fig. 4a and b for different sets of Hamiltonian parameters. The CP² skyrmions are metastable

solutions in the QPM phase for $D \gtrsim 14$, implying that these exotic magnetic-nematic textures should emerge in real magnets under quite general conditions.

Similarly, the metastable pseudo-spin single-skyrmion solutions of the FP phase ($\tilde{\mathcal{B}} > \tilde{\mathcal{B}}_{\text{sat}}$) lead to a spin texture with a nematic (non-magnetic) core and a magnetic (FP) periphery, like the one shown in Fig. 4c. Interestingly, this exotic CP² skyrmion solution remains metastable down to $D \simeq 4|J_1|$ and it coexists with regular (CP¹) metastable skyrmion solutions, like the one shown Fig. 4d, that emerge below $D \simeq 4.25|J_1|$.

## Discussion

We have demonstrated that CP² skyrmion textures emerge in realistic models of hexagonal magnets out of the combination of competing exchange interactions and single-ion anisotropy. It is important to note that the skyrmion crystals and metastable solutions reported in this work survive in the long wavelength limit[26], implying that the CP² skyrmion phases described here should also exist in extensions of the model to honeycomb and Kagome lattice geometries.

There are a number of candidate materials that are well described by the spin-1 model given in Eq. (1). In particular, one may point to the series of triangular antiferromagnets of the form of ABX₃, BX₂, and ABO₂[44,55,56], where A is an alkali metal, B is a transition metal, and X is a halogen atom. Compounds, such as FeI₂[57,58], are described by the Hamiltonian of Eq. (1), but the sign of the single-ion and exchange anisotropies is opposite to the case of interest in this work. Related compounds, such as CsFeCl₃, are known to be quantum paramagnets described by the same Hamiltonian with a dominant easy-plane single-ion anisotropy $D/J_1 \simeq 10$[59]. An alternative route to finding realizations of our spin-1 Hamiltonian is to consider hexagonal materials comprising $4f$ magnetic ions with a singlet single-ion ground state and an excited Ising-like doublet (the effective easy-plane single-ion anisotropy $D$ is equal to the singlet-doublet gap). Ultracold atoms are also powerful platforms to realize spin-1 models with *tunable* single-ion anisotropy[60].

While a full examination of the new response functions and functionalities of the CP² skyrmions must be left to future research, a few remarks should be made here. It is clear that the intrinsically inhomogeneous nature of the local order parameter, which evolves from dipolar to quadrupolar upon moving toward or away from the skyrmion core, can lead to new behaviors. For instance, metastable CP² skyrmions above the saturation field can become stable (ground state) solutions by increasing the $D$ term of a given magnetic ion. This can be achieved with the insertion of non-magnetic impurities that modify the local crystal field. Correspondingly, it should be possible to induce metastable CP² skyrmions by dynamically varying the local crystal field that determines the value of $D$. Furthermore, CP² skyrmions can be manipulated by applying a local strain due to the characteristically non-uniform distribution of the magnitude of their quadrupolar moment.

Before concluding, we remark on a subtle mathematical point. By definition, CP² skyrmions are distinguished from the more familiar CP¹ skyrmions by their enlarged target manifold. This distinction can be physically relevant: a CP² skyrmion will typically have a combination of dipolar and quadrupolar structures. The presence of quadrupole degrees of freedom will bring additional dynamical modes and will have entropic consequences. From a topological perspective, however, there is a certain sense in which CP¹ and CP² skyrmions are equivalent. To elaborate on this point, we first remark that CP¹ is a submanifold of CP². Further, any CP¹ skyrmion can be faithfully embedded in the space of CP² skyrmions, and this embedding preserves skyrmion winding number. Such an embedded spin texture can be smoothly deformed to any other CP² skyrmion with an equal winding number, which establishes a topological equivalence.

In summary, this paper demonstrates that novel magnetic field-induced CP² skyrmion crystals should emerge in the presence of

competing ferromagnetic and antiferromagnetic exchange interactions, a moderate easy-axis exchange anisotropy $\Delta > 2$, and a dominant single-ion easy-plane anisotropy $D$ that is strong enough to stabilize a QPM at $T = 0$. The field-induced quantum phase transition between the uniform quadrupolar state induced by the strong single-ion anisotropy and the $CP^2$ skyrmion crystal is presaged by the emergence of metastable $CP^2$ single-skyrmion solutions exhibiting a magnetic skyrmion core that decays continuously into a quadrupolar periphery. These novel skyrmions can be induced by applying a sufficiently large magnetic field to quantum paramagnets with competing exchange interactions and they can be manipulated with local strain.

The general principles discussed in this work can be generalized to $N$-level systems to obtain $CP^{N-1}$ skyrmion crystals solutions from realistic spin Hamiltonians, illustrating the rich diversity of topological textures that can emerge in magnetic materials due to the quantum mechanical nature of their magnetic moments.

## Methods

The numerical minimization for the phase diagram Fig. 1 is done in a cell of $10 \times 10$ spins containing four magnetic unit cells ($L = 5$). Two crucial steps are useful to improve the efficiency of the local gradient-based minimization algorithms[61]. In the first step, we set multiple random initial conditions $|\mathbf{Z}\rangle$ (-50 for our case), where $|\mathbf{Z}_j\rangle$ on every site $j$ is uniformly sampled on the $CP^2 \simeq S^5/S^1$ manifold. After running the minimization algorithm, we keep the solution with the lowest energy for a given set of Hamiltonian parameters. In the next step, half of the initial conditions are randomly generated, while the other half corresponds to the lowest-energy solutions obtained in the first step within a predefined neighborhood of the Hamiltonian parameters. This procedure is iterated until the phase diagram converges.

## Data availability

All data presented in this study can be reproduced using the code package described in the section "Code availability".

## Code availability

The algorithms used in our numerical simulations are described in the "Methods" section. The numerical code is implemented in Julia and can be found at https://github.com/Hao-Phys/CP2Skyrmions.jl.

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

## Acknowledgements

We acknowledge useful discussions with Xiaojian Bai, Antia Botana, Ying Wai Li, Shizeng Lin, Cole Miles, Martin Mourigal, Sakib Matin, Matthew Wilson, and Shang-Shun Zhang. D. D., K. B. and C.D.B. acknowledge support from the U.S. Department of Energy, Office of Science, Office of Basic Energy Sciences, under Award No. DE-SC0022311. The work by H.Z. was supported by the Graduate Advancement, Training and Education (GATE) fellowship. Z.W. was supported by the U.S. Department of Energy through the University of Minnesota Center for Quantum Materials under Award No. DE-SC-0016371.

## Author contributions

C.D.B. conceived the project. H.Z. and Z.W. designed the numerical code. H.Z. performed the numerical simulations. H.Z. and C.D.B. analyzed the simulation results. D.D. and K.B. performed additional simulations to verify the main conclusions. All authors contributed to the writing of the manuscripts.

## Competing interests

The authors declare no competing interests.
