## [Peer Review File · Nature Communications]

Reviewers' Comments:

Reviewer #1:

Remarks to the Author:

The authors consider a model of an anisotropic frustrated spin-1 magnet on a triangular lattice in an applied magnetic field. The ground states of this model, found using the mean-field approach for spin-coherent states at the magnetic sites, show one or more periodic modulations of three components of the average spin and five components of the quadrupolar order. The authors find crystals of unconventional skyrmions that are quadrupolar-like in the center and spin-like at the periphery and vice versa. I find these results interesting, but I doubt that they are correct, as detailed below.

1. There is no evidence that the inhomogeneous multiply-periodic orders discussed in the manuscript are the actual ground states of the quantum $S = 1$ model Eq.(1). Essentially, the authors solved a classical CP2 model, similar to the one discussed in Ref. [36], the novel element being competing exchange interactions and magnetic anisotropies. There is no small parameter that would justify the classical approach to this $S = 1$ system and the states described in the present manuscript can be destroyed by quantum spin fluctuations: in contrast to the coherent states of harmonic oscillator, the spin fluctuations in the spin-coherent states are not small, for $S = 1$. In particular, it is unclear whether the skyrmion crystal is robust to quantum fluctuations and is actually a ground state of the quantum model.

2. Competing interactions in quantum magnets do not necessarily lead to periodic modulations. They can stabilize valence-bond solids with quantum spin-singlet states formed on bonds or plaquettes (see e.g. M. Sadrzadeh et al., Phys. Rev. B 94, 214419 (2016), M. Mambrini et al., Phys. Rev. B 74, 144422 (2006)) or RVB spin liquids, which cannot be described using on-site spin-coherent states.

3. My understanding is that the actual phase diagram of this $S = 1$ model can only be found by a quantum Monte-Carlo simulation (see e.g. K. Harada et al. Phys. Rev. B 65, 052403 (2002)). Such a calculation would be very tough, in particular, due to frustration, but I do not see how one can get a reliable result otherwise. Perhaps, it can be performed for a single point on the phase diagram, where the authors expect to find the skyrmion crystal state.

4. A model of quantum $S = 1$ spins allows for higher-order interactions, such as $(S_1 S_2)^2$ (see e.g. I Affleck J. Phys.: Condens. Matter 1, 3047 (1989), B. A. Ivanov et al. Phys. Rev. B 68, 052401 (2003)) and their anisotropic variants, which can strongly affect the quadrupolar order and topological defects.

To summarize, I find the results reported in the manuscript interesting and I believe that, if skyrmions will ever be found in quantum $S = 1$ frustrated magnets, they can have a more complex structure than their counterparts in classical spin models, as discussed by the authors. I think, however, that the method used to obtain these results is inadequate for the quantum spin model and for that reason I cannot recommend this manuscript for publication in Nature Communications.

Reviewer #2:

Remarks to the Author:

This article is interesting attempt to find out real models with topological solitons with complicated variables, belonging to that for SU(3) group, within spin systems beyond standard classical representation of spins like classical unit vectors. But the presentation of the results obtained is completely unsatisfactory, either in the "magnetic" part or "mathematical" part of the manuscript. Here are my main remarks.

1. The authors claim the it is hard to realize the Heisenberg model with Dzyaloshinskii-Moriya interaction (DMI). I cannot agree with this statement. It is well-known that non-small MDI appears for thin films (that is just the case of interest for 2D structures) of any magnet grown on the metallic substrate with high spin-orbit interaction, see, e.g., [1*]. Skyrmions and spirals caused by

such DMI are well known. Thus, the presence of DMI is an unavoidable property of nanofilms of magnets and is of interest for "quantum" magnets as well.

2. Note one more material, nickel fluorosilicate $\text{NiSiF}_6 \cdot 6\text{H}_2\text{O}$, for which single-ion anisotropy can be controlled by pressure that leads to transition to quantum paramagnetic state [2*]. By the way, a simple and transparent model for the description of non-linear $\text{SU}(3)$ -spin dynamics was proposed for this material [3*].

3. The authors discuss the only magnets, where quantum effects, i.e., transition from $\text{SU}(2)$ to $\text{SU}(3)$ models, are caused by strong uniaxial anisotropy, whereas many of cited field-theoretical models are based on bilinear-biquadratic spin Hamiltonians. Note that these BL-BQ models are preferable for soliton theory because they can be isotropic and allow continual description. But, of course, spin nematic state differs from quantum paramagnetic state considered here.

4. The CP2 topological solitons cannot exist, strictly speaking, because CP2 is 4D manifold and any mapping of 2D xy coordinate space (S^2 sphere) to 4D is topologically trivial. CP2 is a direct product of two CP1, and the only way to get some topology is to use mapping $xy \rightarrow \text{CP1}$. The authors wrote "Unlike the "usual" CP1 magnetic skyrmions, the dipolar field of the metastable CP2 skyrmions of quantum paramagnets vanishes away from the skyrmion core." The soliton solutions with vanishing dipolar field, i.e., non-zero mean value of spin, were already constructed, see [4*-7*]; these articles should be discussed here.

5. The authors discussed two semiclassical theories with two different sets of variables, for 3D complex unit vector \mathbf{Z} and for 8D real vector \mathbf{n} . For the first case, 4D manifold appears, and for the second one, 6-dimensional. Which is used here? If it is only one, why the other is discussed?

6. Some minor remark:

a) The notation ∇^2 in Eq. (13) and (21) should be explained.

b) The authors wrote about higher spins (probably, $S=3/2$ and $S=2$), but do nothing for these spins. For them, some different technique should be used, see, e.g., [8*] for $3/2$, where 3D complex projective space CP3 is the parameter space. By the way, solitons for quantum spin $3/2$ were discussed in Ref. [4*].

c) The authors wrote "Magnetic skyrmions are nanoscale topological textures that have been recently observed in different families of quantum magnets. These textures are known as CP1 skyrmions". The observations I know were done for standard "classical" magnets, described by Landau-Lifshitz phenomenology.

To conclude, the only novelty I found here is the analysis of highly-anisotropic quantum magnets and predictions of some spiral and soliton solutions. But the model is not purely explained and the comparison with the results obtained before is uncomplete. Thus, the manuscript cannot be published in the present form, and the major revision is necessary.

[1*] A. Fert, Mater. Sci. Forum 59–60, 439 (1990).

[2*] V. G. Bar'yakhtar et al., Sov. Phys. JETP 57, 628 (1983).

[3*] E.G. Galkina et al., Low Temp. Phys. 40, 635 (2014)

[4*] E. G. Galkina et al., Low Temp. Phys. 41, 382 (2015)

[5*] B. A. Ivanov, JETP Letters 84, 84 (2006)

[6*] B. A. Ivanov et al., Phys. Rev. Lett. 100, 047203 (2008).

[7*] H. T. Ueda et al., PRA 93, 021606(R) (2016)

[8*] Y. A. Fridman et al., Phys. Rev. Lett. 106, 097202 (2011).

Reviewer #3:

Remarks to the Author:

[Main results] One of the main results in this manuscript is to find stable CP2 skyrmions crystal and metastable isolated skyrmions in realistic two-dimensional spin-1 models. Another important result is the finding of the metastable CP2 skyrmion against the background quantum paramagnetic state.

[Validity] The authors derived their phase diagram by minimizing the expectation value of the Hamiltonian (1) with the $\text{SU}(3)$ coherent state. The validity of this approximation needs to be explicitly discussed.

The authors claim that "A(s) we will see later, the relevant qualitative aspects of the phase diagram do not depend on the particular choice of the model" in page 3. However, the reviewer

does not find their corroborative reasonings in their manuscript. Particularly, the authors tune the ratio J_1/J_2 (=nearest neighbor exchange energy/ second nearest neighbor exchange energy) so that the magnitude of the ordering vectors is $|\mathbf{b}_1|/5$ and it suffices to calculate the ground state energy in a finite-size system with 10×10 spins. However, in real materials, the ordering vector should be incommensurate in general. Thus, the reviewer wonders how the fine-tuning of J_1/J_2 can be justified when the phase diagram in real materials is discussed.

[Significance]

The stable CP2 skyrmion crystal and metastable in realistic models are new findings in this manuscript because an earlier study [36] (Akagi et al) did not address this point. The importance of their results lies in the relevance to real materials. Actually, the authors raise candidate materials for this theoretical prediction to be confirmed in Sec. V Discussion. Thus the results in this manuscript are important to experimentalists of quantum magnets as well as condensed matter theoretical physicists.

However, the novel responses and functionalities in CP2 skyrmion, which CP1 skyrmions do not have, are needed to be discussed so that significance of the authors' findings becomes clearer.

[Presentation]The manuscripts in the main text and supplemental material are well written and earlier references are appropriately cited. However, let the reviewer point out two small things. (1)"Bar q" in the caption in Fig.2 is not defined. Figure 2 is referred to earlier than Fig.1.

Reviewer 1

The first reviewer's concerns center on the question of whether a semi-classical treatment is appropriate for the model discussed in the manuscript. While we do not agree with the reviewer's objection, we appreciate the opportunity to clarify the issue. As a general response, we have added the following paragraph to the Model section.

“ To study the properties of the skyrmion solutions of Eq. (1), it is first necessary to consider the classical limit. In doing so, we follow the existing literature on topological solitons, which are inherently classical objects. There is, moreover, good reason to expect that quantum fluctuations are not relevant to the present study. Experimentally, there is existing evidence of spin-1 triangular materials that exhibit semi-classical spiral orderings due to competing ferromagnetic and antiferromagnetic exchange interactions [44]. Some of these are discussed in Section V. Furthermore, the results that will be developed will remain unaltered for a simple 3D extension of the current model, achieved by vertically stacking triangular layers with ferromagnetic interlayer coupling. The larger coordination number of the 3D model and the long wave length nature of the ordered states both act to reduce quantum fluctuations, further justifying the classical approximation”.

We address the reviewer's specific comments below.

1. There is no evidence that the inhomogeneous multiply-periodic orders discussed in the manuscript are the actual ground states of the quantum $S = 1$ model Eq.(1). Essentially, the authors solved a classical CP2 model, similar to the one discussed in Ref. [36], the novel element being competing exchange interactions and magnetic anisotropies. There is no small parameter that would justify the classical approach to this $S = 1$ system and the states described in the present manuscript can be destroyed by quantum spin fluctuations: in contrast to the coherent states of harmonic oscillator, the spin fluctuations in the spin-coherent states are not small, for $S = 1$. In particular, it is unclear whether the skyrmion crystal is robust to quantum fluctuations and is actually a ground state of the quantum model.

As a general point, the lack of a small $1/S$ parameter per se does not invalidate the classical approximation. S is not the only possible control parameter. The inverse of the dimension D and the inverse of the wavelength of the magnetic ordering under consideration are additional control parameters that justify the classical approximation. Experimental evidence has demonstrated the validity of such approximations in a very wide range of cases. The success of semi-classical approaches as applied to weakly-frustrated $S = 1/2$ quantum mechanical systems is well documented. It is also well-known that the ground state and the elementary excitations of the nearest-neighbor $S = 1/2$ Heisenberg model on square and honeycomb lattices are well approximated by a semiclassical treatment. Furthermore, the classical approximation extends very

successfully to the three-dimensional case. Indeed, the ground states of the vast majority of known quantum magnets are well approximated by semi-classical theories.

More specifically, there are two arguments to support the classical approach for skyrmion models such as the one considered in our manuscript. The first is that these systems are nearly ferromagnetic in the long-wavelength limit, implying that quantum fluctuations induced by the transverse terms of the Heisenberg interaction become very weak. The second is that the classical phase diagram of the 2D model remains exactly the same as that of a simple 3D extension of the model, achieved by including ferromagnetic exchange between vertically stacked layers. The semiclassical approach is well-known to be robust for 3D systems.

We wish to emphasize that, both theoretically and experimentally, skyrmions are essentially classical objects. Nearly all the successful work that has been done over the past 14 years on magnetic skyrmions has been based on classical treatments of spin Hamiltonians. In particular, the work by Leonov and Mostovoy (Nature Communications volume 6, Article number: 8275 (2015)), which showed that field-induced skyrmion lattices can emerge in centro-symmetric materials, is based entirely on a classical approximation. This work has already accumulated more than 300 citations because of its large impact on our understanding of the mechanisms that can stabilize magnetic skyrmion crystals. Had the referee's criteria been applied to this important work, it would never have been published. We add that Leonov and Mostovoy discuss several spin one quantum magnets (Ni-based materials) as potential realizations of skyrmions crystals as reported in their classical phase diagram.

2. Competing interactions in quantum magnets do not necessarily lead to periodic modulations. They can stabilize valence-bond solids with quantum spin-singlet states formed on bonds or plaquettes (see e.g. M. Sadrzadeh et al., Phys. Rev. B 94, 214419 (2016), M. Mambrini et al., Phys. Rev. B 74, 144422 (2006)) or RVB spin liquids, which cannot be described using on-site spin-coherent states.

The referee is certainly correct that, when special conditions are satisfied, such orderings are possible. These conditions generally include the combination of substantial frustration, low-dimensionality and small spin. However, these conditions are not satisfied here. The frustration is weak. The dominant interaction is nearest-neighbor FM exchange. The competing AFM exchange serves only the purpose of stabilizing long wavelength helical ordering in the absence of field and anisotropy. Additionally, the model is not inherently low-dimensional. Indeed, the model can be extended into three dimensions without changing the phase diagram, as indicated above.

Skyrmion crystals have been reported in multiple magnets, the phase diagrams of which are successfully described by classical theories. For instance, the spin-1/2 magnet Cu_2OSeO_3 includes a skyrmion phase in its phase diagram (DOI: 10.1126/science.1214143). Moreover, the full field temperature phase diagram is quantitatively reproduced by a classical theory. The same is true for countless of spin-1/2 and spin-1 materials whose magnetic orderings and low-energy excitations are well described by coherent states and the corresponding Gaussian fluctuations. Indeed, it is actually very difficult to find quasi-2D or 3D materials that can provide realizations of the phases mentioned by the referee.

3. My understanding is that the actual phase diagram of this $S = 1$ model can only be found by a quantum Monte-Carlo simulation (see e.g. K. Harada et al. Phys. Rev. B 65, 052403 (2002)). Such a calculation would be very tough, in particular, due to frustration, but I do not see how one can get a reliable result otherwise. Perhaps, it can be performed for a single point on the phase diagram, where the authors expect to find the skyrmion crystal state.

While a quantum Monte-Carlo approach would be an appropriate technique to study the more exotic orderings mentioned by the referee in the previous comment, we disagree with the assertion that the zero temperature phase diagrams of all $S = 1$ models can only be validated with QMC. As we have emphasized above, the vast majority of the known $S = 1$ materials exhibit semi-classical orderings. This is the case, for instance, for the Ni-based materials listed at the end of our manuscript.

It should be noted that the study of Harada et al. reports that the QMC phase diagram of the 2D spin one bilinear-biquadratic model studied in that manuscript agrees with the semiclassical theory at zero temperature, which is the case that is considered here as well.

4. A model of quantum $S = 1$ spins allows for higher-order interactions, such as $(S_1 S_2)^2$ (see e.g. I Affleck J. Phys.: Condens. Matter 1, 3047 (1989), B. A. Ivanov et al. Phys. Rev. B 68, 052401 (2003)) and their anisotropic variants, which can strongly affect the quadrupolar order and topological defects.

This is certainly true. However, we are specifically considering Hamiltonians without biquadratic terms precisely because biquadratic and higher order interactions are much smaller than bilinear interactions for the vast majority of known magnets. The reason is that, when derived from Hubbard-like models, these terms are of order t^4/U^3 , while bilinear interactions are of order t^2/U , where t is the typical hopping amplitude and U is the Hubbard on-site repulsion.

The absence of biquadratic terms is in fact what makes the proposed model realistic in a condensed matter context. Unlike strong biquadratic interactions, anisotropic exchange and single site anisotropy are commonly found in real magnetic materials.

To clarify this point, we have added the following sentence in the introduction along with the relevant references: “Biquadratic interactions are typically much smaller than bilinear interactions because they are of higher order in the small parameter (e.g., ratio t/U between the typical hopping amplitude t and on-site Hubbard repulsion for the case of Mott insulators) that leads to the emergence of magnetic moments (localized electrons) in real materials”.

To summarize, I find the results reported in the manuscript interesting and I believe that, if skyrmions will ever be found in quantum $S = 1$ frustrated magnets, they can have a more complex structure than their counterparts in classical spin models, as discussed by the authors. I think, however, that the method used to obtain these results is inadequate for the quantum spin model and for that reason I cannot recommend this manuscript for publication in Nature Communications.

We are glad that the referee is able to see the significance of the more complex skyrmion structures that are possible when $S > 1/2$. We regret that they do not think the classical approach is appropriate for their study, but we feel their objections on this point are misguided.

The referee's central concern seems to be that we have not accounted for the possibility that the proposed Hamiltonian may support exotic and essentially quantum features that cannot be studied with a classical approach. As we have emphasized throughout this response, experimental evidence overwhelmingly supports the idea that the majority of real magnets can be successfully described by a classical theory, whereas the sorts of states described by the referee are notoriously difficult to observe. For the case of magnetic skyrmions in particular, classical treatments have been enormously successful. In this context, the Hamiltonian that we have proposed is in many ways quite ordinary – indeed, that's the point. The character of the exchange is predominantly ferromagnetic, and the frustration is rather weak. It is a natural candidate for a classical treatment. Even under these apparently ordinary conditions, however, skyrmions may emerge that might otherwise be overlooked. This opens up new territory in the study of magnetic skyrmions and suggests to experimentalists that they may find novel skyrmions in materials they might otherwise have dismissed.

Reviewer 2

This article is interesting attempt to find out real models with topological solitons with complicated variables, belonging to that for SU(3) group, within spin systems beyond standard classical representation of spins like classical unit vectors. But the presentation of the results obtained is completely unsatisfactory, either in the “magnetic” part or “mathematical” part of the manuscript. Here are my main remarks.

We thank the referee for recognizing main contribution of our manuscript. We admit that we were not aware of the interesting and relevant literature that was provided by the referee. We now cite this work in the revised manuscript. We have also clarified the issues related to the mathematical parts of the manuscript. Please see detailed responses below.

1. The authors claim the it is hard to realize the Heisenberg model with Dzyaloshinskii-Moriya interaction (DMI). I cannot agree with this statement. It is well-known that non-small MDI appears for thin films (that is just the case of interest for 2D structures) of any magnet grown on the metallic substrate with high spin-orbit interaction, see, e.g., [1*]. Skyrmions and spirals caused by such DMI are well known. Thus, the presence of DMI is an unavoidable property of nanofilms of magnets and is of interest for “quantum” magnets as well.

This seems to be a misinterpretation of the sentence that we wrote in reference to a recent work by Akagi et al (Ref. [39] of the manuscript). To be clear, we were not referring to DMI; rather, we were referring to the SU(3) Heisenberg model that

is considered in Ref. [39]. The SU(3) Heisenberg model is not a realistic model for real magnets because the SU(3) invariance can only be accidental in real magnets and requires an unrealistically large biquadratic interaction. There is no example in the existing literature of a single spin one magnet that is described by such an SU(3) Heisenberg model. In contrast, the model considered in our work does not assume such a high-symmetry in the exchange interaction. It only includes a U(1) invariant XXZ interaction and a single-ion anisotropy term. Hence, we claim that the mechanism to realize CP² skyrmion crystals in quantum magnets provided in our work is much more realistic. To clarify this point, the introduction now contains the following text:

“However, it may be challenging to find materials described by this model because SU(3) can only be an accidental symmetry of the spin-spin interactions of real quantum magnets, and Hamiltonians that do exhibit SU(3)-invariance contain unrealistically strong biquadratic terms. Biquadratic interactions are typically much smaller than bilinear interactions because they are of higher order in the small parameter that leads to the emergence of magnetic moments (localized electrons) in real materials (e.g., the ratio t/U between the typical hopping amplitude, t , and the on-site Hubbard repulsion, U , in the case of Mott insulators). Similar limitations apply to other works that study skyrmion solutions of the bilinear-biquadratic spin one model [40–43]. ”

2. Note one more material, nickel fluorosilicate NiSiF₆H₂O, for which single-ion anisotropy can be controlled by pressure that leads to transition to quantum paramagnetic state [2*]. By the way, a simple and transparent model for the description of non-linear SU(3)-spin dynamics was proposed for this material [3*].

We thank the reviewer for making us aware of this interesting material. There are indeed several materials that can be driven into a quantum paramagnetic state under pressure and an even larger number of spin one materials that are quantum paramagnets at ambient pressure. All of them can be described with a generalized spin wave theory based on SU(3) coherent states. We now cite reference [3*] when we introduce the classical limit based on coherent states of SU(3).

3. The authors discuss the only magnets, where quantum effects, i.e., transition from SU(2) to SU(3) models, are caused by strong uniaxial anisotropy, whereas many of cited field-theoretical models are based on bilinear-biquadratic spin Hamiltonians. Note that these BL-BQ models are preferable for soliton theory because they can be isotropic and allow continual description. But, of course, spin nematic state differs from quantum paramagnetic state considered here.

While we agree on that a biquadratic interaction is preferable for soliton theory, the basic problem, that we point out in the introduction of our manuscript, is that biquadratic interactions are usually very small in real magnets. The simple reason is that, when the Heisenberg model is derived from a Hubbard model, biquadratic interactions appear to order t^4/U^3 , while bilinear interactions are of order t^2/U . Therefore, works based on SU(3) Heisenberg models or bilinear-biquadratic interactions of comparable strength are mainly of academic interest in condensed matter physics. The central point of our

work is that the spin-1 model does not need to be close to an $SU(3)$ invariant point (like the BL-BQ models) to support CP^2 SkXs. Finally, the model that we are proposing also allows for a continuous description [see Eq. (14)] which corresponds an anisotropic CP^2 non-linear sigma model.

Text addressing this issue has been added in the introduction and is reproduced above in the response to the second referee's first question.

4. The CP^2 topological solitons cannot exist, strictly speaking, because CP^2 is 4D manifold and any mapping of 2D xy coordinate space (S^2 sphere) to 4D is topologically trivial. CP^2 is a direct product of two CP^1 , and the only way to get some topology is to use mapping $xy \rightarrow CP^1$.

We agree that, in a strict sense, this is true. The mapping could be written $xy \rightarrow CP^1 \subset CP^2$, where the CP^1 submanifold is generated by an $SU(2)$ subalgebra of $SU(3)$. However, the nontriviality of this mapping is a reflection of the fact that $\pi_2(CP^2) = \mathbb{Z}$. Moreover, the name “ CP^2 Skyrmion” has already been used in the literature (see Ref. [39]).

We wish to emphasize that the essential difference between a CP^1 and a CP^2 skyrmion (in the sense intended here) is not so much mathematical as it is physical. As the referee has suggested, the image of the mapping is indeed isomorphic to CP^1 , however, the CP^1 manifold is here embedded in CP^2 in a physically non-trivial way. Considering the target CP^1 manifold as a Bloch sphere, a traditional CP^1 skyrmion is a texture of dipole moments that point in one direction on the north pole and in the opposite direction on the south pole. For a “ CP^2 skyrmion”, the Bloch sphere is now embedded in CP^2 in a nontrivial way such that points on the sphere no longer correspond exclusively to the dipolar local order parameter. In the example offered in the paper, the north pole corresponds a dipolar order parameter and the south to a quadrupolar one. In a general basis, the mathematical transformation that connects the north and south pole will be an $SU(3)$ transformation rather than an $SU(2)$ rotation.

In our revisions, we have retained the term “ CP^2 skyrmion” in order to be consistent with the literature known to us. However, we are open to revising it if the referee has a reasonable suggestion or can provide a citation with alternate terminology.

We have also added the following clarification in the new version of Sec. V:

“Before concluding, we remark on a subtle mathematical point. By definition, CP^2 skyrmions are distinguished from the more familiar CP^1 skyrmions by their enlarged target manifold. This distinction can be physically relevant: a CP^2 skyrmion will typically have a combination of dipolar and quadrupolar structure. The presence of quadrupole degrees of freedom will bring additional dynamical modes, and will have entropic consequences. From a topological perspective, however, there is a certain sense in which CP^1 and CP^2 skyrmions are equivalent. To elaborate on this point, we first remark that CP^1 is a submanifold of CP^2 . Further, any CP^1 skyrmion can be faithfully embedded in the space of CP^2 skyrmions, and this embedding preserves skyrmion winding number. Such an embedded spin texture can be smoothly deformed to any other CP^2 skyrmion with equal winding number, which establishes a topological equivalence.”

The authors wrote “Unlike the “usual” CP^1 magnetic skyrmions, the dipolar field of the metastable CP^2 skyrmions of quantum paramagnets vanishes away from the skyrmion core.” The soliton solutions with vanishing dipolar field, i.e., non-zero mean value of spin, were already constructed, see [4*-7*]; these articles should be discussed here.

We thank the reviewer for pointing us to these relevant references and have added citations in the new version of our manuscript. We note that these works are similar in spirit to the work by Akagi et al [Ref. 39 of the manuscript] in the sense that the spin models under consideration are not realistic for magnetic materials. While we appreciate that natural deformations of a CP^2 non-linear sigma model, such as the bilinear-biquadratic model, can support these solitons, the actual challenge in condensed matter physics is to find these solutions in realistic models that can guide the search for real materials that will host these topological solitons.

5. The authors discussed two semiclassical theories with two different sets of variables, for 3D complex unit vector Z and for 8D real vector n . For the first case, 4D manifold appears, and for the second one, 6-dimensional. Which is used here? If it is only one, why the other is discussed?

We thank the reviewer for asking this question. The manuscript only presents one semi-classical theory based on coherent states of $SU(3)$. The operator representation of the color field $\mathbf{n}_j = |Z_j\rangle\langle Z_j| - \mathbb{1}/3$ and the coherent states $|Z_j\rangle$ provide equivalent representations of the same classical state [see Refs. 53 and 54 of the manuscript]. The classical phase space manifold is 4-dimensional (dimension of coset $SU(3)/U(2)$, where $U(2)$ is the isotropic group for degenerate irreps of $SU(3)$ [see Ref. 44 of the manuscript]). In other words, both the coherent states and the 8-component real vector belong to a 4D space because they are parametrized by 4 real variables. To emphasize this point, we have added the following to the end of section II:

“We emphasize that the color field formalism just discussed is fully equivalent to the formalism based on coherent states. In particular, it is straightforward to show that the operator representation of the color field may be expressed as $\mathbf{n}_j = |Z_j\rangle\langle Z_j| - \mathbb{1}/3$, in which form it becomes clear that both \mathbf{n} and the coherent state $|Z_j\rangle$ provide equivalent representations of the same classical state [55, 56].”

6. Some minor remark: a) The notation ∇^2 in Eq. (13) and (21) should be explained.

We thank the reviewer for pointing out this potential ambiguity. We now explain that the symbol refers to the spatial gradient operator.

b) The authors wrote about higher spins (probably, $S=3/2$ and $S=2$), but do nothing for these spins. For them, some different technique should be used, see, e.g., [8*] for $3/2$, where 3D complex projective space CP^3 is the parameter space. By the way, solitons for quantum spin $3/2$ were discussed in Ref. [4*].

The theoretical technique is actually very similar. The only difference for CP^{N-1} case is that the classical limit of the theory must be taken based on coherent states of $SU(N)$ [see for instance Phys. Rev. B 106, 054423 (2022)]. The phases of stable and metastable skyrmion solutions are then obtained by minimizing the classical energy over the $2(N-1)$ dimensional CP^{N-1} manifold. Once again, in contrast to the above-mentioned references, the stabilization mechanism does not involve biquadratic interactions. It is important to emphasize this point because the values of the biquadratic interactions (relative to the bilinear exchange) assumed in those works are notoriously difficult to find in real materials. CP^{N-1} skyrmion solutions are well documented in high-energy physics. However, these solutions cannot be “directly transferred” to condensed matter physics because the corresponding models are simply not realistic. A crucial aspect of the present work is to provide similar solutions for much more realistic models.

c) The authors wrote “Magnetic skyrmions are nanoscale topological textures that have been recently observed in different families of quantum magnets. These textures are known as CP1 skyrmions”. The observations I know were done for standard “classical” magnets, described by Landau-Lifshitz phenomenology.

In general, topological solitons are discussed in the the context of classical theories. Nevertheless, one must keep in mind that these classical theories are derived from a quantum mechanical Hamiltonian (as it is described in the manuscript). The actual materials that exhibit magnetic skyrmions are quantum magnets (finite spin systems). The classical Landau-Lifshitz phenomenology is an approximation (classical limit of the quantum mechanical Hamiltonian) that is justified under certain conditions (see our response to the first referee and the new paragraph in the discussions section). Starting from the classical limit described in the manuscript, one can include quantum corrections by quantizing the normal modes (RPA) and implementing a loop expansion (generalization of the $1/S$ expansion for $SU(2)$ coherent states).

To conclude, the only novelty I found here is the analysis of highly-anisotropic quantum magnets and predictions of some spiral and soliton solutions. But the model is not purely explained and the comparison with the results obtained before is incomplete. Thus, the manuscript cannot be published in the present form, and the major revision is necessary.

We thank the reviewer for pointing out these problems with the presentation of the original version of the manuscript. We have clarified both points in the new version of the paper. We also thank the reviewer for recognizing the novelty of our contribution. However, we respectfully disagree with the dismissive tone of the sentence “the only novelty” because our contribution is crucial to find these soliton solutions in real materials. More concretely, in quantum paramagnets, which are usually regarded as uninteresting. The value of this contribution should not be underestimated because finding new magnetic textures in real materials is one the most pressing challenges of modern quantum magnetism. We note that similar guiding principles have led to the experimental discovery of CP^1 magnetic baby skyrmions in 2009 and this discovery

opened a new field in the study of magnetic systems. This is the main reason why we believe that our work should be published in a high-profile journal, such as Nature Communications.

- 1* A. Fert, Mater. Sci. Forum 59–60, 439 (1990).
- 2* V. G. Bar'yakhtar et al., Sov. Phys. JETP 57, 628 (1983).
- 3* E.G. Galkina et al., Low Temp. Phys. 40, 635 (2014)
- 4* E. G. Galkina et al., Low Temp. Phys. 41, 382 (2015)
- 5* B. A. Ivanov, JETP Letters 84, 84 (2006)
- 6* B. A. Ivanov et al., Phys. Rev. Lett. 100, 047203 (2008).
- 7* H. T. Ueda et al., PRA 93, 021606(R) (2016)
- 8* Y. A. Fridman et al., Phys. Rev. Lett. 106, 097202 (2011).

1 Reviewer 3

[Main results] One of the main results in this manuscript is to find stable CP2 skyrmions crystal and metastable isolated skyrmions in realistic two-dimensional spin-1 models. Another important result is the finding of the metastable CP2 skyrmion against the background quantum paramagnetic state.

We thank the reviewer for summarizing the significance and the importance of our work.

[Validity]The authors derived their phase diagram by minimizing the expectation value of the Hamiltonian (1) with the SU(3) coherent state. The validity of this approximation needs to be explicitly discussed.

The authors claim that “A(s) we will see later, the relevant qualitative aspects of the phase diagram do not depend on the particular choice of the model” in page 3. However, the reviewer does not find their corroborative reasonings in their manuscript. Particularly, the authors tune the ratio $J1/J2$ (=nearest neighbor exchange energy/ second nearest neighbor exchange energy) so that the magnitude of the ordering vectors is $|b_1|/5$ and it suffices to calculate the ground state energy in a finite-size system with 10×10 spins. However, in real materials, the ordering vector should be incommensurate in general. Thus, the reviewer wonders how the fine-tuning of $J1/J2$ can be justified when the phase diagram in real materials is discussed.

We thank the reviewer for asking this important question. Actually, the reasoning is provided just above Eq.(21), i.e., when we take the long wavelength limit of the effective low-energy theory, which is valid in the large- D limit. However, we admit that this point was not clearly explained in the original version of the manuscript. The bottom line is that the phase diagram of such theory is known [Refs. 25, 27] and it has been demonstrated to be universal (independent of microscopic parameters) in the long wavelength limit (see Ref. [26]), i.e., near the Lifshitz point $J_2/J_1 = -1/3$. In Sec. IV, we showed that in the large- D limit, the spin-one model Eq. (1) reduces to an effective pseudo spin-1/2 model Eq. (19). It has also been established that the qualitative aspects of the phase diagram remain insensitive to the ratio of J_1/J_2 for wavelengths as short as five lattice spaces. This means that the ratio J_2/J_1 can vary over a significant range ($2/(1+\sqrt{5}) \gtrsim |J_2|/J_1 > 1/3$) without changing the qualitative aspects of the phase diagram. We have added a few lines above Eq.(21) to further clarify this important point.

[Significance] The stable CP2 skyrmion crystal and metastable in realistic models are new findings in this manuscript because an earlier study [36] (Akagi et al) did not address this point. The importance of their results lies in the relevance to real materials. Actually, the authors raise candidate materials for this theoretical prediction to be confirmed in Sec. V Discussion. Thus the results in this manuscript are important to experimentalists of quantum magnets as well as condensed matter theoretical physicists.

We thank the reviewer for identifying the significance of our work.

However, the novel responses and functionalities in CP2 skyrmion, which CP1 skyrmions do not have, are needed to be discussed so that significance of the authors' findings becomes clearer.

We thank the referee for asking this question. In first place, we believe that the main contribution of the work is to point out that field-induced CP2 magnetic skyrmions and skyrmion crystals can emerge in *quantum paramagnets*. So the main significance of our work lies in providing guiding principles to find skyrmions in a new class of non-magnetic materials that are usually regarded as uninteresting. That said, we agree with the referee on that it is also relevant to point out the new responses and functionalities that are expected for CP2 skyrmions. While this should be the subject of future research, it is clear that the intrinsic inhomogeneous nature of the local order parameter, which evolves from dipolar to quadrupolar upon moving towards or away from the skyrmion core can lead to novel effects. For instance, metastable CP² skyrmions above the saturation field can become stable (ground state) solutions by locally increasing the D term via the insertion of non-magnetic impurities that modify the crystal field. Correspondingly, it should be possible to induce metastable CP² skyrmions by applying a pulse of strain that locally modifies the value of D . More generally, we expect strong pinning effects induced by inhomogeneous strain that couples directly to the quadrupolar region of the CP² skyrmions. A new paragraph along these lines has been added in the "Discussion" section of the revised version of the manuscript.

[Presentation]The manuscripts in the main text and supplemental material are well written and earlier references are appropriately cited. However, let the reviewer point out two small things. (1)“Bar q” in the caption in Fig.2 is not defined. Figure 2 is referred to earlier than Fig.1. We need to think about the novel responses and functionalities in CP2 skyrmions.

We thank the reviewer for pointing out these problems. We have made corresponding changes in our manuscript.

Reviewers' Comments:

Reviewer #1:

Remarks to the Author:

Replying to my comment on the importance of quantum effects in $S = 1$ systems with unconventional magnetic orders, the authors argued that quantum spin fluctuations are suppressed if the wavelength of modulated spin structures, such as skyrmion crystals, is large and the interlayer coupling is taken into account. This argument is correct, but it pertains to a small vicinity of the Lifshitz point, at which the modulation vector vanishes. The region occupied by skyrmion crystals is already small in the magnetic field vs single-ion anisotropy phase diagram (Fig. 1) and the chances to find skyrmion crystals would be even smaller, if the fine tuning of the exchange constants ratio, J_2/J_1 , is required. In absence of the fine tuning, skyrmion radius would be comparable with the lattice constant. Such small skyrmions (narrow skyrmion tubes in three dimensions) can be destroyed by quantum spin fluctuations.

Overall, I am satisfied with the authors' reply to my and other referees' comments. This manuscript reports an interesting theoretical prediction and can be published in Nature Communications.

I have one additional comment concerning the candidate materials discussed in the discussion section. I do not believe that the magnetic anisotropy of octahedrally coordinated Ni^{2+} ions (Refs. [59-61]) can greatly exceed the strength of exchange interactions, which seems to be required for stability of skyrmion crystals (Figure 1 in the present manuscript). The reason is the quenched orbital moment of the e_g electrons. The authors might consider mentioning that skyrmions are unlikely to be found in these materials. Compounds with the Ising-like Fe^{2+} (Co^{2+}) ions or strongly anisotropic rare earth ions, mentioned by the authors, are a more likely playground for discovering the states predicted in this manuscript.

Reviewer #2:

Remarks to the Author:

The authors made a big work to improve their manuscript in line with my comments. I think, the revised version of the manuscript can be published in Nature Communications as it is now.

Reviewer #3:

Remarks to the Author:

Considering the revised manuscript and the authors' reply, I would like to comment further on validity and importance.

My previous comment on validity:

[Validity]The authors derived their phase diagram by minimizing the expectation value of the Hamiltonian (1) with the SU(3) coherent state. The validity of this approximation needs to be explicitly discussed. The authors claim that "A(s) we will see later, the relevant qualitative aspects of the phase diagram do not depend on the particular choice of the model" in page 3. However, the reviewer does not find their corroborative reasonings in their manuscript. Particularly, the authors tune the ratio J_1/J_2 (=nearest neighbor exchange energy/ second nearest neighbor exchange energy) so that the magnitude of the ordering vectors is $|b_1|/5$ and it suffices to calculate the ground state energy in a finite-size system with 10×10 spins. However, in real materials, the ordering vector should be incommensurate in general. Thus, the reviewer wonders how the finetuning of J_1/J_2 can be justified when the phase diagram in real materials is discussed.

Authors' reply:

We thank the reviewer for asking this important question. Actually, the reasoning is provided just above Eq.(21), i.e., when we take the long wavelength limit of the effective low-energy theory, which is valid in the large- D limit. However, we admit that this point was not clearly explained in the original version of the manuscript. The bottom line is that the phase diagram of such theory is

known [Refs. 25, 27] and it has been demonstrated to be universal (independent of microscopic parameters) in the long wavelength limit (see Ref. [26]), i.e., near the Lifshitz point $J_2/J_1 = -1/3$. In Sec. IV, we showed that in the large- D limit, the spin-one model Eq. (1) reduces to an effective pseudo spin-1/2 model Eq. (19). It has also been established that the qualitative aspects of the phase diagram remain insensitive to the ratio of J_1/J_2 for wavelengths as short as five lattice spaces. This means that the ratio J_2/J_1 can vary over a significant range ($2/(1 + \sqrt{5}) \gtrsim |J_2/J_1| > 1/3$) without changing the qualitative aspects of the phase diagram. We have added a few lines above Eq.(21) to further clarify this important point.

My second comment on validity:

I thank the authors' explanation and revision. However, I am not yet convinced by the authors' claim that the ratio J_2/J_1 can vary over a significant range ($2/(1 + \sqrt{5}) \gtrsim |J_2/J_1| > 1/3$) without changing the qualitative aspects of the phase diagram in their reply.

Their claim relies on refs. 25-27. I could suggest the authors to present their reasoning more explicitly because I do not find it in a specific way.

(As far as I understand, in refs 26 and 27, models on triangular lattice in the large Q case have been studied only for particular values of $J_3/J_1=0.5$ and $J_3 \sim 1.62$ (Fig8 in ref.27).

I infer that their claim is based on comparison between Fig. 7 for a range of A and Fig. 8(b) for $T=0$. Is it correct?)

Accordingly, numerical results on fine structure of phase diagram presented in the inset of Fig.1 on case with large Q is better to be presented in more precise way; the phase boundary is drawn by the straight lines but numerical results should be a finite number of dots like Fig.7 in ref. 26. Further h dependence of the energy per site of different states for a particular value of D should be presented (e.g. in the supplemental material) in a way similar to Fig. 2 in ref. 27 so that the validity of those results becomes clearer to the readership .

I agree on the existence of Skyrmion crystal in the large D limit in the long wave length (i.e. small Q)limit. However, this holds when Δ is larger than two so that \tilde{D} in (22) is positive. This condition is better to be mentioned.

My previous comment on importance:

...However, the novel responses and functionalities in CP2 skyrmion, which CP1 skyrmions do not have, are needed to be discussed so that significance of the authors' findings becomes clearer.

Authors' reply:

We thank the referee for asking this question. In first place, we believe that the main contribution of the work is to point out that field-induced CP2 magnetic skyrmions and skyrmion crystals can emerge in quantum paramagnets. So the main significance of our work lies in providing guiding principles to find skyrmions in a new class of nonmagnetic materials that are usually regarded as uninteresting. That said, we agree with the referee on that it is also relevant to point out the new responses and functionalities that are expected for CP2 skyrmions. While this should be the subject of future research, it is clear that the intrinsic inhomogeneous nature of the local order parameter, which evolves from dipolar to quadrupolar upon moving towards or away from the skyrmion core can lead to novel effects. For instance, metastable CP2 skyrmions above the saturation field can become stable (ground state) solutions by locally increasing the D term via the insertion of non-magnetic impurities that modify the crystal field. Correspondingly, it should be possible to induce metastable CP2 skyrmions by applying a pulse of strain that locally modifies the value of D . More generally, we expect strong pinning effects induced by inhomogeneous strain that couples directly to the quadrupolar region of the CP2 skyrmions. A new paragraph along these lines has been added in the "Discussion" section of the revised version of the manuscript.

My second comment on importance:

I am afraid that the authors' reply is not satisfactory for me; I agree with the authors that a guiding principle of new states of matter is as itself important. However, this manuscript should remark (at least briefly) or appeal how fascinating the new topological matter is.

Reviewer 1

I have one additional comment concerning the candidate materials discussed in the discussion section. I do not believe that the magnetic anisotropy of octahedrally coordinated Ni^{2+} ions (Refs. [59-61]) can greatly exceed the strength of exchange interactions, which seems to be required for stability of skyrmion crystals (Figure 1 in the present manuscript). The reason is the quenched orbital moment of the e_g electrons. The authors might consider mentioning that skyrmions are unlikely to be found in these materials. Compounds with the Ising-like Fe^{2+} (Co^{2+}) ions or strongly anisotropic rare earth ions, mentioned by the authors, are a more likely playground for discovering the states predicted in this manuscript.

We thank the reviewer for the useful observation. We agree with the reviewer on that the magnetic *exchange* anisotropy Δ of octahedrally coordinated Ni^{2+} ions (Refs. [59-61] of the previous version of the manuscript) is not expected to be significantly larger than one for magnetic ions whose orbital magnetic moment is quenched. We have then removed Refs. [59-61] as the candidate materials to realize our model in the new manuscript.

Reviewer 2

The authors made a big work to improve their manuscript in line with my comments. I think, the revised version of the manuscript can be published in Nature Communications as it is now.

We thank the reviewer for the constructive criticisms that helped us to improve the new version of the manuscript and for recommending the manuscript for publication.

Reviewer 3

I thank the authors' explanation and revision. However, I am not yet convinced by the authors' claim that the ratio J_2/J_1 can vary over a significant range $2/(1+\sqrt{5}) \gtrsim |J_2|/J_1 > 1/3$ without changing the qualitative aspects of the phase diagram in their reply. Their claim relies on refs. 25-27. I could suggest the authors to present their reasoning more explicitly because I do not find it in a specific way. (As far as I understand, in refs 26 and 27, models on triangular lattice in the large Q case have been studied only for particular values of $J_3/J_1 = 0.5$ and $J_3 \approx 1.62$ (Fig8 in ref.27). I infer that their claim is based on comparison between Fig. 7 for a range of A and Fig. 8(b) for T=0. Is it correct?)

As pointed out by the reviewer, Refs. [25, 27] studied the frustrated models on the triangular lattice for fixed values of J_2/J_1 and J_3/J_1 . However, Ref. 26 is a Ginzburg-Landau analysis of these models in the *long wavelength limit*. The crucial observation

is that the universal $T = 0$ phase diagram that is obtained in the long wavelength limit (see Fig. 7 of Ref. 26) is qualitatively the same as the phase diagrams [Fig. 1 of Ref. 25 and Fig. 3(a) of Ref. 27] that were obtained for fixed $J_{2,3}/J_1$. In particular, for a fixed intermediate value of the single-ion easy-axis anisotropy A , both phase diagrams include a single- Q vertical spiral phase, a skyrmion crystal phase, and a fully polarized phase as a function of increasing magnetic field h . As we have shown in the manuscript, the spin-one Hamiltonian reduces to an effective pseudo-spin-1/2 Hamiltonian (see Eq. (19)) in the large- $D/|J_1|$ limit. Moreover, Eq. (19) takes the form given in Eq. (21) in the long-wavelength limit, which is the same as the *universal* Hamiltonian Eq. (37) of Ref. 26. Consequently, *in the large- D limit* and for an intermediate value of \tilde{D} [see Eq. (22)], the $T = 0$ phase diagram must also include three phases {MVS-II, SkX-II, and FP (pseudo-spin “up”)} for $\tilde{h} > 0$ and {QPM (pseudo-spin “down”), SkX-I, MVS-I} for $\tilde{h} < 0$. This is indeed the case for our phase diagram obtained for $|Q| = 2\pi/5$, which is shown in the inset of Fig. 1. The comparison is indeed based on Fig. 7 for a range of A and Fig. 8(b) for $T = 0$.

We have expanded the explanation above Eq. (21) to make these points more explicit.

Accordingly, numerical results on fine structure of phase diagram presented in the inset of Fig.1 on case with large Q is better to be presented in more precise way; the phase boundary is drawn by the straight lines but numerical results should be a finite number of dots like Fig.7 in ref. 26. Further h dependence of the energy per site of different states for a particular value of D should be presented (e.g. in the supplemental material) in a way similar to Fig. 2 in ref. 27 so that the validity of those results becomes clearer to the readership.

We thank the reviewer for allowing us to demonstrate the numerical accuracy our results. We have included a new section in the Supplementary Information that explains the careful procedure that we used to determine the phase boundaries. As requested by the reviewer, the new figure (Fig. S4) shows the magnetic field dependence of the energy per site and the magnetization ($D/|J_1| = 19$). As it is clear from the new figure, our determination of the phase boundaries are very accurate due to the efficient gradient-based algorithm that is explained in **Methods**. This is the reason why we can use lines to indicate the phase boundaries. Note also that the use of lines (instead of points) for variational $T = 0$ phase diagrams is a standard procedure in the field (see for instance Fig. 1 of Ref. 25).

I agree on the existence of Skyrmion crystal in the large D limit in the long wave length (i.e. small Q) limit. However, this holds when Δ is larger than two so that \tilde{D} in (22) is positive. This condition is better to be mentioned.

We thank the reviewer for pointing this out. We have made this point more explicit in the last paragraph of Sec. V (“... skyrmion crystals should emerge in the presence of competing ferromagnetic and antiferromagnetic exchange interactions, **a moderate easy-axis exchange anisotropy** $\Delta > 2$, and a dominant single-ion easy-plane anisotropy $D \dots$ ”).

I am afraid that the authors' reply is not satisfactory for me; I agree with the authors that a guiding principle of new states of matter is as itself important. However, this manuscript should remark (at least briefly) or appeal how fascinating the new topological matter is.

We have added a few lines to the concluding paragraphs emphasizing the rich diversity of topological textures that can emerge in quantum magnets.

Reviewers' Comments:

Reviewer #3:

Remarks to the Author:

The authors have respond all suggestions in my second report and revised accordingly. Consequently, validity of their results, reasoning and importance are more clarified. In conclusion, the revised manuscript deserves a publication of this journal.

REVIEWERS' COMMENTS

Reviewer #3 (Remarks to the Author):

The authors have respond all suggestions in my second report and revised accordingly. Consequently, validity of their results, reasoning and importance are more clarified. In conclusion, the revised manuscript deserves a publication of this journal.

We reply:

We thank the reviewer for the constructive comments.